# Effects of Dandelion Extract on Promoting Production Performance and Reducing Mammary Oxidative Stress in Dairy Cows Fed High-Concentrate Diet

**DOI:** 10.3390/ijms25116075

**Published:** 2024-05-31

**Authors:** Yan Zhang, Musa Mgeni, Ziqing Xiu, Yu Chen, Juncai Chen, Yawang Sun

**Affiliations:** College of Animal Science and Technology, Southwest University, Chongqing 400716, China; zy051305@email.swu.edu.cn (Y.Z.); xzq1023@email.swu.edu.cn (Z.X.);

**Keywords:** dairy cows, high-concentrate diet, LPS, dandelion extract, oxidative stress

## Abstract

This study investigated the effects of rumen bypass dandelion extract on the lactation performance, immune index, and mammary oxidative stress of lactating dairy cows fed a high-concentrate diet. This study used a complete randomized block design, and initial milk production, somatic cell counts, and parities were set as block factors. Sixty Holstein cows with similar health conditions and lactating periods (70 ± 15 d) were divided into three groups with 20 replicates per group. The treatments included the LCD group (low-concentrate diet, concentrate–forage = 4:6), HCD group (high-concentrate group, concentrate–forage = 6:4), and DAE group (dandelion aqueous extract group, HCD group with 0.5% DAE). The experimental period was 35 d, and cows were fed three times in the morning, afternoon, and night with free access to water. The results showed the following: (1) Milk production in the HCD and DAE groups was significantly higher (*p* < 0.05) than that in the LCD group from WK4, and the milk quality differed during the experimental period. (2) The HCD group’s pH values significantly differed (*p <* 0.01) from those of the LCD and DAE groups. (3) In WK2 and WK4 of the experimental period, the somatic cell counts of dairy cows in the HCD group were significantly higher (*p* < 0.05) than those in the DAE group. (4) The serum concentrations of 8-hydroxy-2’-deoxyguanosine (8-OHdG) and protein carbonyl (PC) in the HCD group were significantly higher (*p <* 0.05) than those in the LCD group. The activity of catalase (CAT) in the LCD and DAE groups was stronger (*p <* 0.01) than that in the HCD group. (5) The correlation analysis revealed significantly positive correlations between the plasma LPS concentration and serum concentrations of 8-OHdG (*p <* 0.01), PC (*p <* 0.01), and malondialdehyde (MDA, *p <* 0.05) and significantly negative correlations (*p <* 0.01) between the plasma LPS concentration and activities of CAT and superoxide dismutase. (6) Compared with that in the HCD and DAE groups, the mRNA expression of α, β, and κ casein and acetyl CoA carboxylase in bovine mammary epithelial cells was significantly higher (*p <* 0.05) in the LCD group, and the mRNA expression of fatty acid synthetase and stearoyl CoA desaturase in the LCD group was significantly higher (*p <* 0.01) than that in the HCD group. (7) Compared with that in the LCD and HCD groups, the mRNA expression of Nrf2 was significantly higher (*p <* 0.01) in the DAE group, and the mRNA expression of cystine/glutamate transporter and NAD (P) H quinone oxidoreductase 1 in the DAE group was significantly higher (*p <* 0.05) than that in the HCD group. Overall, feeding a high-concentrate diet could increase the milk yield of dairy cows, but the milk quality, rumen homeostasis, and antioxidative capability were adversely affected. The supplementation of DAE in a high-concentrate diet enhanced antioxidative capability by activating the Nrf2 regulatory factor and improved rumen homeostasis and production performance.

## 1. Introduction

Lactation is a demanding period for the animal, which requires a significant effort of the organism to meet all needs, but also the effort of farmers to provide those needs. Despite the action of homeostatic mechanisms to maintain the nutritional balance, changes in metabolites and hormones occur as a result of increased metabolic demands in lactating animals. These changes make animals physiologically unstable and more susceptible to a number of metabolic diseases compromising productivity as well as welfare [1,2,3]. In order to promote the economic efficiency of livestock farms experiencing a shortage of high-quality forage, breeders have added more concentrate to the diets of dairy cows to ensure that their feed provides enough energy to support high milk production. However, the long-term feeding of a high-concentrate diet can lead to metabolic abnormalities in the rumen, including a reduction in pH caused by the accumulation of short-chain fatty acids [4] and a disrupted rumen equilibrium state, which results in the lysis and death of a large number of cells. Lipopolysaccharide (LPS), the main component of the Gram-negative bacterial cell wall, is then released into the rumen fluid and impairs the barrier function of the bovine gastrointestinal epithelium [5,6,7]. As a consequence, LPS inevitably enters the circulatory system through the gastrointestinal epithelium, leading to an increase in LPS levels in peripheral blood, including the mammary artery and veins [8,9,10,11]. LPS in the mammary gland of cows can stimulate the production of a large amount of reactive oxygen species (ROS), impairing milk quality and disrupting the redox homeostasis in milk [12]. Currently, antimicrobial therapy plays a significant and consistent role in mastitis treatment in dairy cow production industries. However, antimicrobial resistance has become a worldwide public concern, so it is urgent to seek substitutes for antibiotics and reduce antimicrobial usage.

Dandelion (*Taraxacum officinale* (L.) Weber ex F.H. Wigg) is a common perennial herbaceous plant in the Asteraceae family that originated in Europe and is widespread in the southwest and northwest regions of China. It is a non-toxic, potentially antioxidative, anti-rheumatic, and anti-inflammatory herb [13]. There are ancient medical records, from about one thousand years ago, of dandelion being used to treat postpartum breastfeeding mothers with poor milk discharge and other conditions. The bioactive ingredients of dandelion, such as phenolic acids and flavonoids, can effectively inhibit the formation of free radicals and eliminate excessive free radicals in the body [14], which is the basis of its antioxidant properties. One experiment obtained an extract from a dandelion flower using ethyl acetate as a medium and found it to contain high concentrations of xluteolin, luteoside, caffeic acid, and chlorogenic acid, which were found to possess the strongest ability to inhibit hydroxyl radical activity and scavenge activated free radicals [15,16]. It has also previously been observed that dandelion extract can reduce intracellular oxidative reactions and enhance the activities of antioxidant enzymes such as superoxide dismutase in mouse and human colonic epithelial cells that have been subjected to oxidative stress/damage and inflammatory responses induced by perhydrol [17,18,19]. In in vivo experiments, the activity of endogenous antioxidants in mice was significantly improved after they were fed dandelion root and leaf extract [20]. It has previously been observed in feeding trials that dandelion extract can significantly increase the activity of antioxidant enzymes in the plasma of male rabbits and improve the proportional lipid composition. This indicates that dandelion has potential antioxidant and hypolipidemic effects and can also inhibit atherosclerosis induced by oxidative stress [21]. Meanwhile, there is a growing body of literature that recognizes that these active ingredients not only play the role of deoxidizers but also can regulate the expression of antioxidant genes as well as the activity of antioxidant enzymes through activating various signaling pathways. Very little attention has been paid to the effects of dandelion extract on large domestic animals or to statistics regarding its influence on production efficiency in long-term in vivo experiments. Moreover, researchers have not examined the relationship between the LPS content in cows and oxidative stress in the body, including the mammary gland, in detail.

This study set out to investigate whether dandelion extract could improve the production performance of dairy cows fed a high-concentrate diet and alleviate the oxidative stress induced by excessive LPS in the body.

## 2. Results

### 2.1. Effects of Different Diets on Milk Production, Feed Intake, and Milk Synthesis Efficiency of Dairy Cows

Figure 1 shows a clear trend of decreasing milk production in the LCD group at the beginning of the trial due to the increased ratio of forage in the diet. Milk production in the HCD and DAE groups fluctuated during the trial, but the overall level was basically the same as in the pre-trial period. In the third week, the LCD group’s milk production was significantly impaired (*p* < 0.05) compared to that in the HCD group, while the difference between the DAE group and the other two groups was insignificant. From the fourth week onwards, the milk production of cows in the LCD group declined significantly (*p* < 0.05) compared to that in the HCD and DAE groups, and there was no significant difference between the HCD and DAE groups.

As shown in Figure 2, during the trial, the DMI of the LCD group was highly significantly (*p <* 0.01) lower than that of the HCD and DAE groups, whereas there was no significant difference in the DMI of the HCD and DAE groups. The milk synthesis efficiency in the LCD group was significantly (*p <* 0.05) higher than that in the HCD and DAE groups in the first week and increased from week 2 to week 5.

### 2.2. Effects of Different Diets on Milk Somatic Cell Count (SCC) and Milk Quality of Dairy Cows

The somatic cell counts in the three groups in different periods are shown in Figure 3. Compared to that of the other two groups, the SCC of the HCD group was higher and varied less. In contrast, the SCC in the LCD and DAE groups first declined significantly and then gradually stabilized. In the second and fourth weeks, the SCC of the HCD group was significantly higher than that of the DAE group (*p* < 0.05), whereas the LCD group did not differ significantly from the other two groups.

Table 1 illustrates the effects of the three different diets on milk quality. The milk fat contents in the LCD group were higher than those in the HCD group throughout the experiment and showed a significant (*p <* 0.05) difference in the fifth week. No significant differences between the milk fat contents of the DAE group and those of the other groups were found. In the second week, the milk solid non-fat (MSNF) in the DAE significantly (*p <* 0.01) exceeded that in the LCD and HCD groups. Meanwhile, the MSNF contents in the HCD and DAE groups were highly significant (*p <* 0.01) and significantly (*p <* 0.05) higher than those in the LCD group in the third and fourth weeks, respectively.

The milk density in the LCD group was significantly (*p <* 0.05) lower than that in the HCD group in the first week and declined significantly more (*p <* 0.05) than that in the DAE group in the second week. In the third week, the milk density of the DAE group significantly increased (*p <* 0.01) compared to that of the HCD group and was significantly higher (*p <* 0.01) than that of the LCD group. In the fourth week, the milk density was significantly (*p <* 0.01) higher in the HCD and DAE groups than in the LCD group. The protein content in the DAE group significantly (*p <* 0.01) exceeded that in the LCD and HCD groups in the third week. The milk lactose content of the HCD and DAE groups was significantly (*p <* 0.01) higher than that in the LCD group in the third and fourth weeks.

### 2.3. Effects of Different Diets on Rumen pH, LPS Concentration in the Rumen Fluid, and Plasma of Dairy Cows

The rumen pH in cows fed a high-concentrate diet was significantly lower (*p <* 0.01) compared to that in the cows fed a low-concentrate diet (Figure 4A), whereas, when the dandelion extract was added to the high-concentrate ration, the rumen pH significantly increased (*p <* 0.01). With the addition of dandelion extract to the ration, the rumen fluid content of LPS was significantly higher (*p <* 0.01) than that in the LCD group and also significantly dropped (*p <* 0.01) compared to that in the HCD group (Figure 4B). The plasma content of LPS in the cows in the HCD and DAE groups was significantly higher (*p <* 0.01) than that in the LCD group (Figure 4C).

### 2.4. Effects of Different Diets on the Concentration of Oxidative Damage Markers and Antioxidant Enzyme Activities in Serum

As shown in Figure 5, the contents of three oxidative damage markers in the sera of cows in the HCD group were higher than those in the LCD and DAE groups. Among them, the concentrations of 8-hydroxy-2′-deoxyguanosine (8-OHdG) and protein carbonyl (PC) in the HCD group were significantly higher (*p <* 0.05) than those in the LCD group. Figure 6 shows that, under different treatments, the serum catalase (CAT) activity was more significantly (*p <* 0.01) reduced in the HCD group than in the LCD and DAE groups. The differences in the activities of other antioxidant enzymes are not significant.

### 2.5. Analysis of Correlation between Plasma LPS Concentration and Oxidative Damage Markers and Antioxidant Enzyme Activities

There was a significant positive correlation (Y = 138.798 + 274.547X, R^2^ = 0.414, *p <* 0.01; Y = 15.665 + 18.479X, R^2^ = 0.233, *p <* 0.01) between the LPS content in the plasma and serum 8-OHdG and PC levels (Figure 7A,B). At the same time, the plasma LPS content and serum malondialdehyde (MDA) (Figure 7C) showed a significant positive correlation (Y = 8.593 + 8.104X, R^2^ = 0.154, *p* = 0.039). As Figure 8 shows, there was a highly significant negative correlation between plasma LPS content and serum CAT activity (Y = 230.406 − 497.342X, R^2^ = 0.431, *p <* 0.01), and a significant negative correlation with serum SOD activity (Y = 10.970 − 11.574X, R^2^ = 0.219, *p* = 0.010). There was no significant correlation between plasma LPS content and serum GSH-Px and T-AOC content.

### 2.6. Effects of Different Diets on mRNA Expression of Genes Related to Lactation

As shown in Figure 9, the mRNA expression of genes related to lactation varied greatly among different dietary groups. The mRNA expression of the genes related to casein synthesis (*CSN1*, *CSN2*, and *CSN3*) in the epithelial cells of mammary glands of cows in the LCD group was significantly higher than that in the HCD and DAE groups. With regard to the genes related to milk fat synthesis, the mRNA expression of fatty acid synthetase (*FASN*) and stearoyl COA desaturase (*SCD*) in the LCD group was significantly higher (*p <* 0.01) than that in the HCD and DAE groups, while the differences compared to the DAE group were insignificant. In addition, the LCD group’s mRNA expression of the CoA carboxylase α (*ACACA*) gene in mammary epithelial cells was significantly higher (*p <* 0.05) than that in the HCD and DAE groups.

### 2.7. Effects of Different Diets on mRNA Expression of Antioxidant-Related Genes

As shown in Figure 10, the mRNA expression of three genes, *Nrf2*, *XCT*, and *NQO-1*, in mammary epithelial cells differed significantly between the groups fed different rations. The mRNA expression of Nrf2 in the DAE group was significantly higher (*p* < 0.01) than that in the LCD and HCD groups. While the mRNA expression of both the *XCT* and *NQO-1* genes was significantly higher (*p <* 0.05) in the DAE group than in the HCD group, the difference compared to the LCD group was not significant.

## 3. Discussion

Milk yield is a comprehensive performance indicator that reflects various aspects of a cow’s body condition and is influenced by many factors, such as breed, lactation stage, litter size, feeding level, nutrient absorption and metabolic rate, and immune status. The results show that, when the concentrate-to-roughage ratio in the LCD group was reduced from 5.2:4.8 to 4:6 in the trial, the milk production gradually decreased, and the gap between the HCD and DAE groups gradually grew. This was closely related to the decrease in energy level in the ration. Zhou et al. [22] show that different concentrate-to-roughage ratios (65:35 and 46:54) had a significant effect on the milk yield of cows; a high-concentrate ration with a net energy content of 1.54 Mcal/kg significantly elevated the milk yield compared to a low-concentrate ration with a net energy content of 1.40 Mcal/kg. Abaker et al. [23] found that, after feeding different concentrate-to-roughage ratios (6:4 and 4:6) for 18 consecutive weeks, the milk yield of cows was significantly higher in the high-concentrate ration group than in the low-concentrate ration group. However, from the ninth week onwards, the milk yield of the high-concentrate group gradually decreased and fell more than that of the low-concentrate group. This was mainly due to the inflammation and oxidative stress in cows caused by long-term feeding with a high-concentrate ration. The milk production of the DAE group was almost equal to that of the HCD group in this trial. However, since few studies have reported the in vivo testing of dandelion and its extracts in dairy cows, the results of this trial can only indicate that dandelion extracts have a small effect on the milk production of dairy cows. Few studies have obtained similar results to the present test after adding flavonoid-rich alfalfa extract [24] and propolis extract [25] to dairy cow rations. Prior studies have shown that, when grape pomace extracts [26], green tea, and curcuma extracts [27] were added to the ration, these flavonoid-rich additives improved milk production in dairy cows. These studies suggest that different sources of flavonoid extracts have different effects on milk production in dairy cows.

Many studies have found that, during the occurrence of SARA in dairy cows, their feed intake is reduced to different levels, and this reduction in dry matter intake (DMI) is a shared symptom of SARA [28]. This is mainly induced by anorexia due to elevated propionic acid levels and blood glucose levels in the rumen [29], as well as high ruminal osmotic pressure, dehydration, and endotoxemia [30,31]. However, the high DMI of the HCD and DAE groups in this trial could be attributed to the shorter feeding period with a high-concentrate ration, which may have failed to elevate propionic acid and glucose levels in the rumen. In addition, the dry matter percentage in the high-concentrate ration at the same volume was significantly higher than that of the low-concentrate ration. However, cows in the LCD group had higher milk production efficiency, although the DMI was significantly decreased compared to that in the other two groups. The excessive production of LPS in organisms under a high-concentrate diet can trigger an inflammatory response in the mammary glands of cows. Therefore, many nutrient precursors are utilized by the immune system to produce acute phase proteins (APP), causing a reduction in nutrients entering the synthesis pathway for milk components.

The somatic cell count (SCC) is an important indicator of milk quality and dairy cow health. Milk SCCs are predominantly leukocytes, usually including macrophages, neutrophils, and lymphocytes, with a small number of detached mammary epithelial cells. Numerous studies have found a strong relationship between significant milk SCC growth and the occurrence of SARA or mastitis. This is mainly due to an altered mammary gland immune health status, which results in many immune cells entering the mammary gland and being secreted into the milk [32,33]. In this trial, all the cows had high SCCs at the beginning because they were fed a ration with a high concentration according to the farm’s needs. However, as the trial progressed, the SCC in the LCD and DAE groups declined, whereas the counts in the HCD group were similar to the pre-trial values. As mentioned in one study [34], flavonoid-rich pomegranate peel extract decreased the milk SCC of cows with different somatic cell bases and lactation periods and improved milk production. In addition, reports have shown that, upon adding different concentrations of alfalfa flavonoids to the rations of dairy cows, a significant reduction in SCC in the milk at an additional level of 60 mg/kg BW was observed. This further shows that flavonoid extracts can reduce the immune cell content in milk and decrease the occurrence of inflammation [24].

Fat, non-fat milk solids, density, protein, and lactose in milk are representative measures of milk quality and important indicators of cow performance. A strong relationship between a reduction in milk fat and protein and the occurrence of SARA [35] was proved early in 1997. The main reason is that SARA elicited by high-concentrate rations causes an exponential increment in LPS levels in the body and an elevated immune activation status. This leads to a redistribution of nutrients both before and after entry to the mammary gland so that more nutrients can go to the immune response and fewer enter the mammary gland for the synthesis of milk components. Zebeli and Ametaj [36] showed that, when increasing the barley content in a cow’s diet, the rumen LPS content and plasma levels of C-reactive protein, serum amyloid A, and lipopolysaccharide-binding protein were significantly enhanced. At the same time, the milk fat, milk yield, and milk energy efficiency significantly decreased. In this experiment, the milk fat in the high-concentrate ration group was decreased compared to that in the low-concentrate ration group. On the contrary, there was more lactose in milk from cows fed the high-concentrate ration than in the low-concentrate group. This accorded with previous observations, which found a growth in milk yield and lactose content but a reduction in milk protein and milk fat after glucose injection via arterial entry. An alternative explanation for this is that lactose synthesis promotes the entry of excess water from the mammary glands into the milk [37]. The DAE had little effect on milk quality and only showed some improvement in milk protein and fat content in the high-concentrate feeding mode. Previous studies have shown that the effects of different sources of flavonoid extracts on the milk quality of dairy cows varied greatly. After feeding green tea and curcuma extract continuously for 12 weeks starting from 3 weeks before parturition [27], the milk fat percentage of dairy cows improved by 10%, and the addition of silymarin and lycopene [38] to the ration from 1 week before to 2 weeks after parturition resulted in a 13% increase in the milk fat. Meanwhile, adding alfalfa brassica extract from about day 80 of lactation did not promote a significant difference in the milk quality of cows [24].

The pH and LPS levels in the rumens of dairy cows are important when evaluating the changes in the rumen environment and the occurrence of rumen acidosis. The concentrate-to-roughage ratio in the high- and low-concentrate groups in this experiment was determined with reference to the experimental design of Khafipour et al. [10] and Abaker et al. [23], whose concentrate-to-roughage ratio of 6:4 triggered the occurrence of SARA in dairy cows [39]. The high-concentrate group showed a significant decrease in rumen pH and a significant boost in rumen and plasma LPS levels compared to the low-concentrate group. The findings of the present study were consistent with those of the above-mentioned experiments, which indicated that the high-concentrate ration disrupted the homeostasis of the rumen’s internal environment and damaged the rumen wall, with an exponential rise in the amount of LPS entering the blood circulation of the cows. Another interesting finding was that the coating treatment of DAE also retarded the rumen acidosis and reduced the LPS content in the rumen and plasma compared with those in the high-concentrate group. There are two types of flavonoids that naturally exist: monomer and complex. While the biological activity of monomeric flavonoids in the rumen is greatly disrupted due to high degradation rates, the biological activity of complex flavonoids in the rumen remains high [40]. To date, only a few studies have focused on the effects of flavonoid-rich additives on the internal rumen environments of dairy cows under a high-concentrate feeding pattern. A study has shown that adding a flavonoid essential oil mixture to the diet of young heifers significantly alleviated the decrease in reticulum pH caused by the high-concentrate ration. At the same time, there was a significant decrease in their rumen LPS content and serum LPS-binding protein (LBP) [41]. Meanwhile, a few studies have suggested that the mechanism by which flavonoids are able to inhibit the decrease in rumen pH is closely related to the alteration of rumen fermentation patterns and an increase in lactic acid consumption due to numerous anaerobic lactic acid-consuming bacteria [42,43].

As the most common pathogen-associated molecular pattern, LPS binds to TLR4 on the cell membrane and activates related transcription factors, such as NF-κB, inducing innate immune responses and further triggering systemic inflammatory responses. Proinflammatory factors, which are produced in large amounts in the inflammatory response, can stimulate intracellular reactive oxygen species (ROS) production. A considerable amount of literature shows that LPS is involved in most of the toxic inflammatory responses regulated by ROS and nitrogen radicals (NOS) [44,45]. Furthermore, higher concentrations of LPS in the systemic circulation directly stimulate the production of ROS by some immune cells, such as neutrophils and koilocytes. The recruitment of such cells at the immune activation site may also trigger localized high ROS content and subsequent oxidative damage [46]. Moreover, in vitro experiments showed that the co-culture of MAC-T cells with LPS-activated neutrophils for 24 h resulted in the release of superoxide and gelatinase from the activated neutrophils and resulted in MAC-T cell damage [47]. It has been demonstrated that the contents of oxidative damage markers and antioxidant enzyme activities in blood and liver tissues changed dramatically after cows consumed high-concentrate rations, accompanied by an increase in LPS content in the body circulation [48,49]. In the present study, ANOVA of the oxidative damage markers between different treatments and analysis of the correlations between plasma LPS and the oxidative damage markers revealed that the contents of DNA and protein oxidative damage markers in the HCD group were significantly higher than those in the LCD group and that there was a significant positive correlation between the LPS level in the plasma and the contents of oxidative damage markers. This further proved that higher concentrations of LPS in the systemic circulation could trigger oxidative damage in the organism. Meanwhile, after analyzing the antioxidant enzyme activity, it was found that CAT activity was significantly reduced in the HCD group compared with the LCD group. Further, the correlation analysis showed that plasma LPS concentration was significantly negatively correlated with the activities of CAT and SOD, which are two of the most important enzymes in the antioxidant defense system. They can catalyze the metabolic decomposition of intracellular superoxide radicals into the less oxidatively active hydrogen peroxide and water [50]. Excessive LPS inhibits the activities of the above two enzymes, thus inactivating the antioxidant defense system in the organism and making the organism more susceptible to oxidative stress.

DAE possesses biological activities such as antioxidant and anti-inflammatory effects due to its rich content of flavonoids and phenolic acids, but most of the research on its antioxidant properties has focused on in vitro cellular assays, while relatively few in vivo experiments have been carried out in animals. Choi et al. [21] added dandelion root and leaf hay powder directly to the diets of rabbits with high cholesterol, which significantly increased the activity of antioxidant enzymes in the plasma and reduced oxidative stress-induced atherosclerosis. When DAE was added to the diet of weaned piglets, it could significantly alleviate the diarrhea caused by weaning stress, increase the rates at which certain nutrients were metabolized, and improve the environment of the intestinal flora. However, DAE had no significant effect on the antioxidant capacity of the piglets [51]. The antioxidant capacity and lipid peroxidative damage were significantly improved in beef cattle [52], sheep [53], and lactating dairy cows [54] when other flavonoid-rich extracts were added to the diet. However, when cows were in the periparturient period, in which they were more sensitive to oxidative stress, the improvement in antioxidant function caused by flavonoid extracts was no longer significant [26,27]. In addition, the addition of quercetin to the ration of dairy cows at the end of lactation had no effect on the abundance of mRNA transcripts because antioxidant enzyme genes have a greater influence on the antioxidant effects of flavonoid extracts in the liver [55]. In this experiment, DAE could alleviate the oxidative damage of DNA, proteins, and lipids caused by the high-concentrate feeding pattern of mid-lactation cows while significantly increasing the activity of CAT and enhancing the antioxidant capacity of the organism.

Casein is a calcium- and phosphorus-containing binding protein and the predominant protein in mammalian milk. Casein accounts for 80% of the total milk protein in cow milk and has three main types: α, β, and κ [56]. After constructing a model of mastitis in dairy cows via the intramammary infusion of Escherichia coli, Gunther et al. [57] found that the synthesis of casein as well as expression of the αS1 casein mRNA in cow’s milk was significantly reduced. Similarly, some studies have also found a significant decrease in the expression of caseins α and β in mammary tissues, as well as a significant down-regulation of the expression of related pathway factors that regulate their synthesis in cows fed high-concentrate diets for a long time. This corresponded to a significant up-regulation of the expression of LPS-binding proteins and inflammatory factor-related genes [54,58]. In addition to milk proteins, the content of milk fat also plays a crucial role in milk quality and energy. Milk fatty acid synthesis occurs via two pathways: One is the de novo synthesis of short- and medium-chain fatty acids in the mammary gland. This process is mainly regulated by the key enzymes Sterol Regulatory Element Binding Protein 1, *ACACA*, and *FASN*. The other pathway is the direct uptake of long-chain fatty acids from the blood. This process is mainly regulated by carriers such as fatty acid-binding protein (*FABP*) and fatty acid transporter protein (*FATP*) [59]. The content and composition of fatty acids in milk are susceptible to various factors, such as breed, stage of lactation, the lipid composition of the ration, and feeding management. In a cell culture study, Chen et al. [60] examined genome-wide methylation and the methylation levels of the promoter regions of lactation-related genes following the LPS treatment of MAC-T cells, and the methylation of genes related to the amino acid, lipid metabolism, and immune response pathways was significantly elevated when the LPS treatment concentration ranged from 1 to 10 EU/mL. This suggested that the reasons for the decrease in milk protein and milk fat synthesis in mammary epithelial cells after LPS treatment include the elevated degree of gene methylation, which inhibits the expression of milk protein synthesis genes. Tian et al. [61] also found a connection between an increase in DNA methylation in the mammary gland and prolonged intake of the high-concentrate diet by lactating goats, as well as a decrease in milk fat content and down-regulation of the expression of genes related to milk synthesis. Recent evidence suggests that milk fat synthesis in dairy cows plummets when SARA occurs. The main reason for this is the sudden decrease in the amount of fatty acids available for milk fat synthesis in the mammary gland [62]. The expression of genes related to milk protein and milk fat synthesis in the mammary epithelial cells of cows in the HCD group was significantly down-regulated compared to that in the LCD group in this experiment, which is in agreement with previous studies. In contrast, in the DAE group, the expression of genes related to milk protein and milk fat synthesis was moderately up-regulated compared to that in the HCD group, but the difference was insignificant. This may be related to the antioxidant and anti-inflammatory properties of the active ingredients in DAE. DeNardi et al. [41] found that blood levels of three acute phase proteins, SAA, LBP, and binding bead protein (Hp), were significantly reduced after the addition of polyphenol-essential oil mixtures to the high-concentrate diets of dairy cows, suggesting that polyphenol-essential oil mixtures significantly improved the immune stress statuses of cows. The data on SCC, blood, and oxidative stress in this trial showed a similar effect of DAE, which in turn led to an increase in nutrient precursors in the mammary gland that can be used for the synthesis of milk lipids and milk proteins.

Nuclear factor erythroid 2-related factor 2, the most widely studied and the most important factor involved in regulating the body’s resistance to oxidative stress, has also been examined in the field related to the resistance to oxidative stress in dairy cows. However, most of these studies have focused on the regulatory role of *Nrf2* in major immune organs such as the liver and systemic immune functions, and only a few studies have noted the presence of *Nrf2* in the mammary tissues of dairy cows in response to oxidative stress. Memon et al. [63] found that LPS could trigger oxidative stress by inducing the activation of the *MAPK* signaling pathway under a high-concentrate feeding pattern and simultaneously inhibiting the *Nrf2* transcriptional regulator to trigger oxidative stress. The addition of polyphenolic linolein to dairy cow rations can improve the body’s antioxidant capacity by up-regulating the expression of mRNA of *Nrf2*-associated genes in the mammary gland [64]. Recent studies have shown that adding methionine to periparturient dairy cow diets significantly promoted the phosphorylation of *Nrf2* in mammary tissues while down-regulating *Keap1* protein expression. This, in turn, up-regulated the expression of *Nrf2*-regulated target genes with antioxidant effects [65]. In this experiment, there were no significant differences in the mRNA expression of genes related to *Nrf2* transcriptional regulators in mammary glands in the high-concentrate ration group. Still, DAE highly significantly increased the mRNA expression of *Nrf2* and significantly increased the mRNA expression of downstream antioxidant genes regulated by *Nrf2*, *XCT*, and *NQO-1*. Upon combining the results from current and previous studies regarding the mechanism of action of DAE, we concluded that DAE can alleviate mammary gland oxidative damage and inhibit oxidative stress caused by high-concentrate rations by activating the *Nrf2* transcription factor and its downstream antioxidant gene expression.

## 4. Materials and Methods

### 4.1. Ethic Statement

Ethical approval for the experiment was obtained from the Animal Ethics Committee of Southwest University of China for animal welfare (Number: 3167130267). The animal feeding and treatment were performed in accordance with relevant requirements.

### 4.2. Animal, Experiment Design, Feeding Management, and DAE

Sixty healthy *Holstein* cows (with close litter size) in the mid-lactation period (70 ± 15 d) were selected based on their milk yield (34.20 ± 0.65 L) and health condition and divided into 3 groups of 20 cows, each using a completely randomized design. Three groups were designed as follows: (1) a low-concentrate diet group (LCD, concentrate–forage = 4:6); (2) a high-concentrate diet group (HCD, concentrate–forage = 6:4); (3) a dandelion aqueous extract group (DAE, HCD group with 0.5% DAE). Next, 500 g feed samples from the three groups were taken on d0, d17, and d35 and stored at −20 °C. We analyzed the feed samples using the method mentioned by Wu (2021), and the ingredients and nutrient levels of the three diets are listed in Table 2.

The feeding period lasted for 42 days, including 7 days of adaptation. Cows were fed at 07:30, 14:30, and 19:30 daily and had free access to water. All the cows were milked three times a day at 06:30, 13:30, and 18:30. The feed intake was recorded once a week by measuring the leftovers of each cow after 1 h of feeding. Dried powder of dandelion extract was purchased from Daosifu Biotechnology Co., Ltd. (Nanjing, Jiangsu, China). The DAE was applied to the rumen protection, undertaken by Hangzhou King Techina Technology Co., Ltd. (Hangzhou, Zhejiang, China), to decelerate its degradation in the rumen. The proportional flavonoid content in DAE was analyzed using ultraviolet–visible (UV–vis) spectrophotometry, and the proportion of flavonoid content in DAE used in this trial was 7%. The contents of Isorhamnetin-3-o-glucoside and quercetin in the DAE were measured using high-performance liquid chromatography (HPLC) with the protocol mentioned in a previous study [15], and the levels were 0.052% and 0.262%, respectively (Appendix A).

### 4.3. Sample Collection, Production Performance, and Isolation of Mammary Epithelial Cells

All the samples were collected from 10 cows from each group (20 cows) with moderate body conditions and milk production. On the last day of the trial, 1 L of intermediate milk was collected from cows in the afternoon and stored in a refrigerator at −20 °C to collect mammary epithelial cells and extract intracellular RNA. The rumen fluid was collected within 2 h after morning feeding on the day before the end of the experiment. The collection tube was soaked in warm water containing 0.1% Neosporin and inserted about 1.5–1.8 m into the mouth. To avoid salivary contamination, the first 30–50 mL of rumen fluid was discarded. Then, 100 mL of rumen fluid was withdrawn and divided into two portions. One was immediately used to test the pH with a portable pH meter, and the other was filtered with four layers of sterile gauze and transferred into a sterile, heat-exempted freezing tube and stored in liquid nitrogen. On day 33, before morning feeding, 5 tubes of blood were collected from each cow: 3 without anticoagulant for serum, 1 with EDTA anticoagulant for whole blood, and 1 with sodium heparin anticoagulant for plasma. Serum and plasma were separated on the day of collection using a refrigerated centrifuge (Eppendorf, 5910R, Hamburg, Germany) at 4 °C and 3500 rpm for 10 min and stored in a refrigerator at −80 °C. Serum was used to measure the content of oxidative damage markers and antioxidant enzyme activities, and plasma was used to measure the content of LPS. After collection, whole blood was stored in a refrigerator at 4 °C and sent to the Animal Hospital of Southwest University the next day for analysis with a Blood Routine Analyzer (Healthy Life, HF-3200, Wuxi, Jiangsu, China). Milk production was recorded once a week, and milk yield was recorded once a week by summing the three milk yields in the morning, mid-day, and evening of the same day. Then, 30 mL of mid-milk was collected in a 50 mL centrifuge tube for each cow once a week in the afternoon. This was used to determine the SCC directly using a milk quality tester (Kepuda Technology Co. Company, NFMT-100, Chengdu, China) and a milk somatic cell detector (Dairy quality, RT10, Ottawa, Canada). The remaining milk samples were stored at −20 °C in a freezer.

### 4.4. Determination of LPS

The LPS levels in the rumen fluid and plasma were detected using an End-point Chromogenic Lyophilized Tachypleus Amebocyte Lysate Kit (Xiamen Bioendo Technology Co., Ltd., Xiamen, China). Before determination, the collected rumen fluid samples were thawed at room temperature and centrifuged at 10,000 rpm for 5 min at 4 °C. The supernatant was then transferred to a 2 mL sterile, chilled centrifuge tube. After thawing at room temperature, the collected plasma centrifugal supernatant was mixed well using a vortex shaker. Then, the processed rumen fluid and plasma samples were used to analyze the LPS content according to the instruction manual of the kit. The optical density (OD) value was read at 405 nm using a microplate reader (Bio-Rad, xMark™, Hercules, CA, USA). The standard curve formula was derived, and the absorbance values of the sample wells were substituted into the formula for calculation.

### 4.5. Determination of Oxidative Damage Markers and Antioxidant Enzyme Activity

The levels of oxidative damage markers (8-OHdG, MDA, and PC) and the activities of antioxidant enzymes (*SOD*, CAT, GSH-Px, and T-AOC) in serum samples were measured using bovine ELISA kits (Mlbio, Shanghai, China). The assay was performed according to the procedure outlined by our previous study [66]. The general procedure was as follows: loading the sample, spiking the sample, adding the enzyme reagent to each well except the blank wells, incubating, washing with washing solution, and then color development. The final samples were used to measure the absorbance of individual wells at 450 nm using an enzyme meter. The final dilution of the sample in this test included 5 iterations.

### 4.6. Isolation of RNA, cDNA Synthesis, and Quantitative Real-Time PCR (qPCR)

RNA was isolated in mammary epithelial cells collected from milk according to the procedure outlined by Wu (2021). The procedure can be summed as follows: 200 mL of the milk sample was sieved through a cell sieve, poured into a 250 mL centrifuge flask, and then centrifuged for 10 min at 3500 rpm at 4 °C. The precipitate was treated with PBS and centrifuged at 4 °C, 3500 rpm for 10 min. The process was repeated to collect the precipitate. The precipitate was again treated with PBS and centrifuged at 4 °C and 3500 rpm for 10 min. Lysate was added, and intracellular RNA was extracted according to the instructions of the FastPure Cell/Tissue Total RNA Isolation Kit purchased from Vazyme Biotech Co., Ltd. (Nanjing, China).

RNA was reverse transcribed following the instructions of the iScript cDNA synthesis Kit (Bio-Rad, Hercules, CA, USA). The qPCR procedure followed that proposed by our previous study [66]. The primers for the antioxidant genes listed in Table 3 were created using Primer Premier 5.0 software. The coSmpany BGI Co., Ltd. (Shenzhen, China) handled the synthesis. The antioxidant genes included *XCT*, *SOD*, *NQO-1*, and *HO-1*. The glyceraldehyde-3-phosphate dehydrogenase gene (*GAPDH*) was employed as a housekeeping gene. The following protocols were used for the amplification and quantification: pre-incubation for 30 s at 95 °C, 40 cycles of denaturation for 5 s at 95 °C, and annealing for 5 s at 60 °C. Then, the 2^−∆∆CT^ method was utilized to measure the relative mRNA expression levels, and the melting curves were examined to verify specific amplification.

### 4.7. Statistical Analysis

The data were presented using means ± standard errors (SEs). All the analyses were conducted using SPSS 19.0 (IBM Inc., Armonk, NY, USA, 2010). One-way analysis of variance followed by Duncan’s multiple comparisons was carried out on each variable to determine the variability between treatments. A *p* value *<* 0.05 was considered significant, and the significance levels were set at 1%.

## 5. Conclusions

In the high-concentrate feeding mode, the milk yield of dairy cows can be maintained at a high level. However, the milk quality, rumen homeostasis, and antioxidant capacity of the body and mammary gland were affected. The addition of DAE enhanced the antioxidant capacity of the mammary gland and the body by activating the *Nrf2* transcriptional regulator in the mammary gland and further improved the lactation performance, as well as significantly increasing the rumen pH and inhibiting the production of LPS.

## Figures and Tables

**Figure 1 ijms-25-06075-f001:**
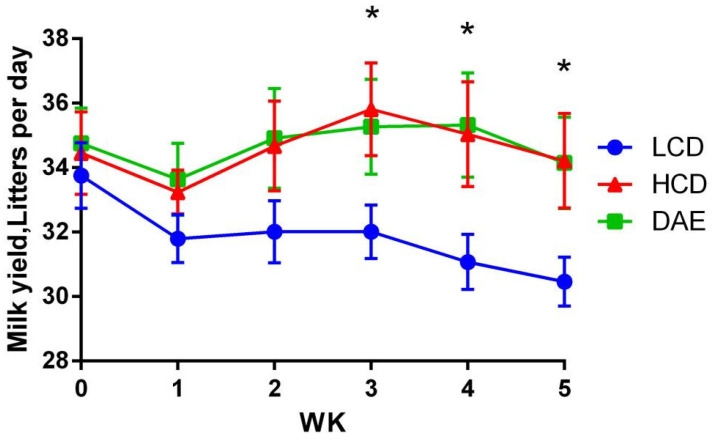
Effects of different diets on milk production of dairy cows. LCD: low-concentrate group, concentrate–forage = 4:6; HCD: high-concentrate group, concentrate–forage = 6:4; DAE: dandelion aqueous extract group, with the addition of 0.5% DAE in HCD diet; * indicates significant difference between treatments (*p <* 0.05).

**Figure 2 ijms-25-06075-f002:**
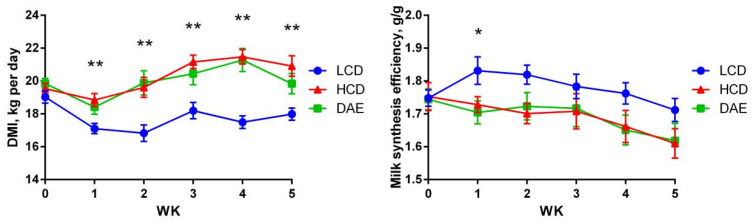
Effects of different diets on dry matter intake (left panel) and milk synthesis efficiency (right panel). LCD: low-concentrate group, concentrate–forage = 4:6; HCD: high-concentrate group, concentrate–forage = 6:4; DAE: dandelion aqueous extract group, with the addition of 0.5% DAE in HCD diet; * indicates significant difference between treatments (*p* < 0.05); ** indicates highly significant difference between treatments (*p* < 0.01).

**Figure 3 ijms-25-06075-f003:**
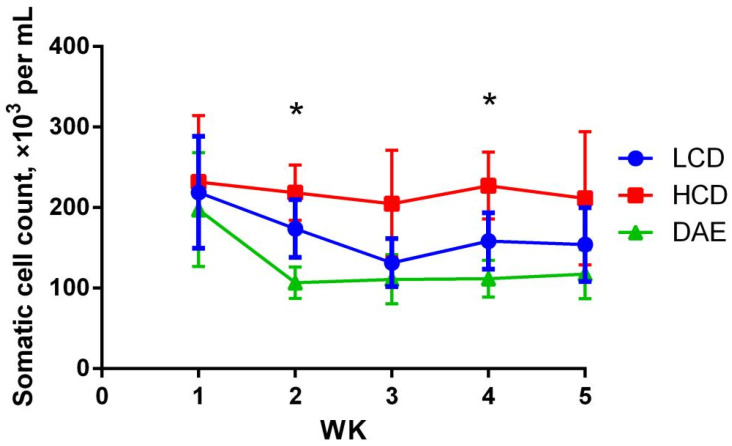
Effects of different diets on SCC in milk. LCD: low-concentrate group, concentrate–forage = 4:6; HCD: high-concentrate group, concentrate–forage = 6:4; DAE: dandelion aqueous extract group, with the addition of 0.5% DAE in HCD diet; * indicates significant difference between treatments (*p <* 0.05).

**Figure 4 ijms-25-06075-f004:**
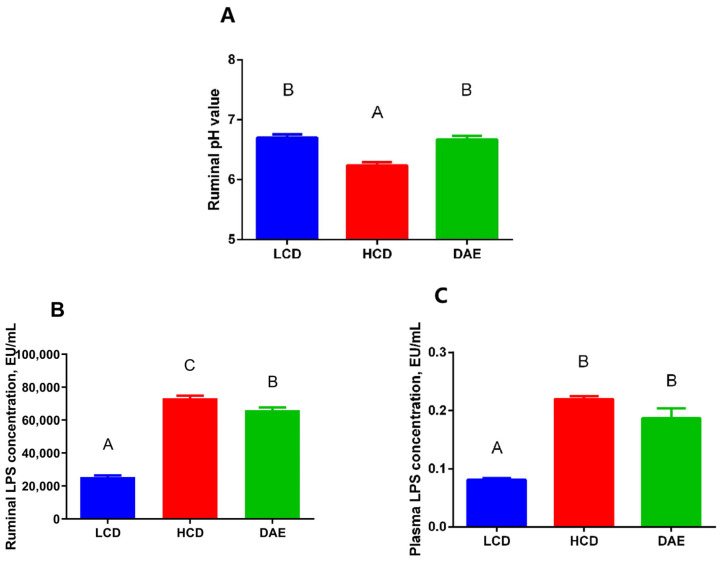
Effects of different diets on the ruminal pH value (**A**), ruminal lipopolysaccharide (LPS) concentration (**B**), and plasma LPS concentration (**C**) in dairy cows. LCD: low-concentrate group, concentrate–forage = 4:6; HCD: high-concentrate group, concentrate–forage = 6:4; DAE: dandelion aqueous extract group, with the addition of 0.5% DAE in HCD diet. Different capital letters (A, B, and C) indicate significant differences between treatments (*p <* 0.01).

**Figure 5 ijms-25-06075-f005:**
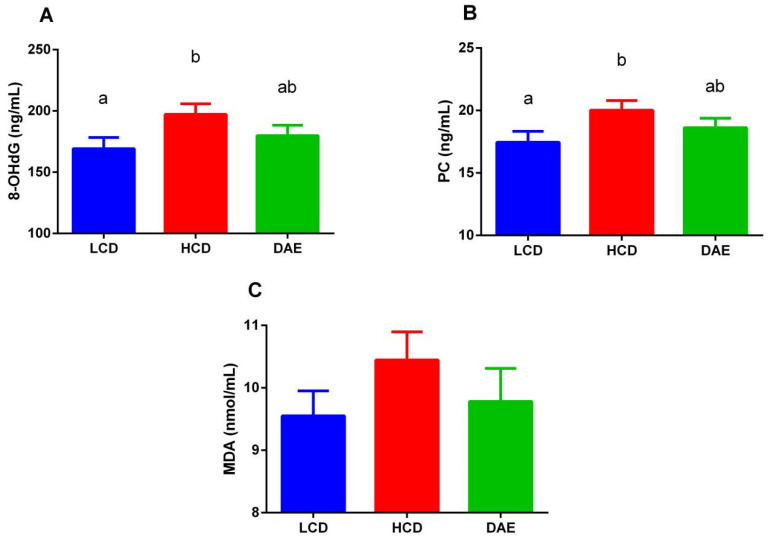
Effects of different diets on oxidative damage markers in the serum of dairy cows, including 8-hydroxy-2′-deoxyguanosine (8-OHdG) (**A**), protein carbonyl (PC) (**B**), and malondialdehyde (MDA) (**C**). LCD: low-concentrate group, in a concentrate-to-roughage ratio of 4:6; LCD: low-concentrate group, concentrate–forage = 4:6; HCD: high-concentrate group, concentrate–forage = 6:4; DAE: dandelion aqueous extract group, with the addition of 0.5% DAE in HCD diet. Different lowercase letters (a and b) indicate significant differences between treatments (*p <* 0.01).

**Figure 6 ijms-25-06075-f006:**
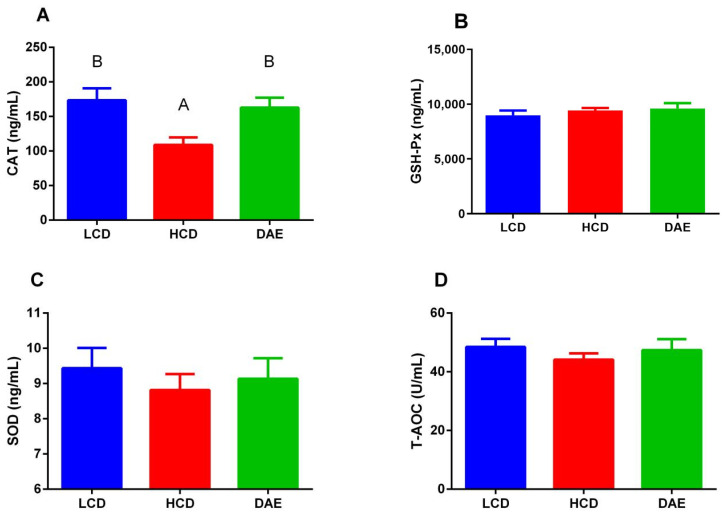
Effects of different diets on oxidative damage markers in the serum of dairy cows, including catalase (CAT) (**A**), glutathione peroxidase (GSH-Px) (**B**), superoxide dismutase (SOD) (**C**), and total antioxidant capacity (T-AOC) (**D**). LCD: low-concentrate group, concentrate–forage = 4:6; HCD: high-concentrate group, concentrate–forage = 6:4; DAE: dandelion aqueous extract group, with the addition of 0.5% DAE in HCD diet. Different capital letters (A and B) indicate significant differences between treatments (*p <* 0.01).

**Figure 7 ijms-25-06075-f007:**
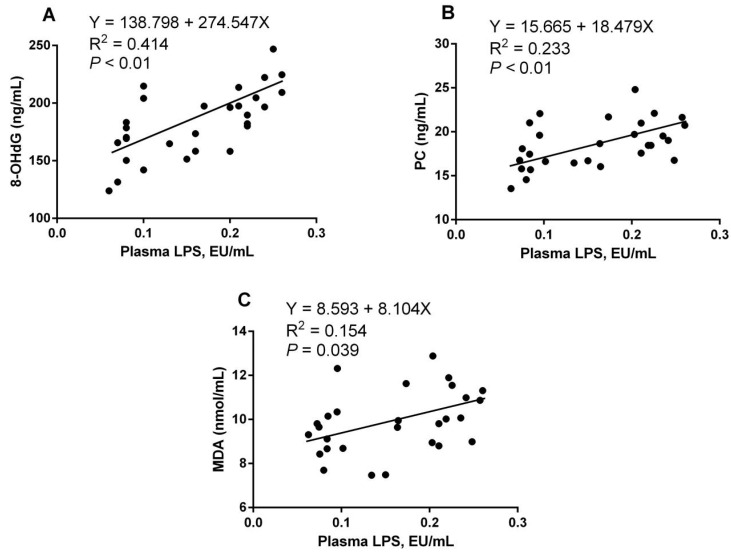
Correlations between the concentrations of plasma lipopolysaccharide (LPS) and oxidative damage markers, including 8-hydroxy-2′-deoxyguanosine (8-OHdG) (**A**), protein carbonyl (PC) (**B**), and malondialdehyde (MDA) (**C**).

**Figure 8 ijms-25-06075-f008:**
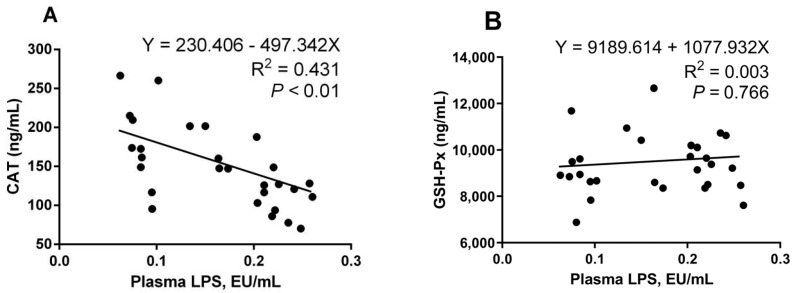
Correlations between concentrations of plasma lipopolysaccharide (LPS) and activities of antioxidative enzymes, including catalase (CAT) (**A**), glutathione peroxidase (GSH-Px) (**B**), superoxide dismutase (SOD) (**C**), and total antioxidant capacity (T-AOC) (**D**).

**Figure 9 ijms-25-06075-f009:**
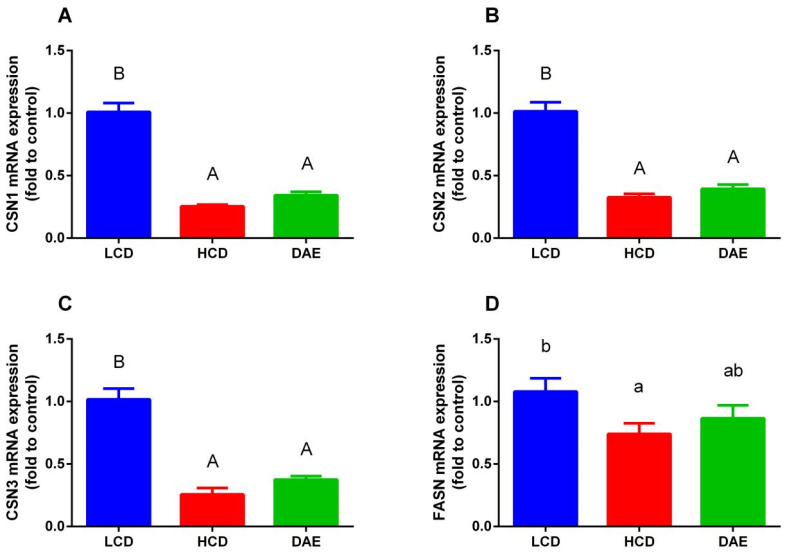
Effects of different diets on mRNA expression of milk synthesis-related genes in dairy cows. (**A**) Casein α (*CSN1*); (**B**) casein β (*CSN2*); (**C**) casein κ (*CSN3*); (**D**) fatty acid synthetase (*FASN*); (**E**) stearoyl COA desaturase (*SCD*); (**F**) acetyl CoA carboxylase α (*ACACA*). LCD: low-concentrate group, concentrate–forage = 4:6; HCD: high-concentrate group, concentrate–forage = 6:4; DAE: dandelion aqueous extract group, with the addition of 0.5% DAE in HCD diet. Different capital letters (A and B) indicate highly significant differences between treatments (*p <* 0.01). Different lowercase letters (a and b) indicate significant differences between treatments.

**Figure 10 ijms-25-06075-f010:**
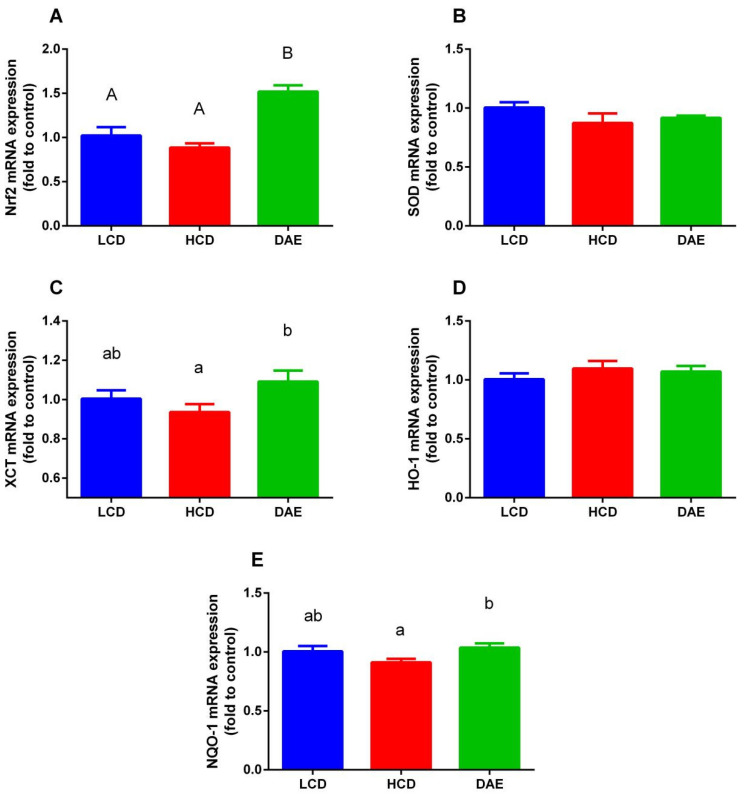
Effects of different diets on mRNA expression of antioxidative genes in dairy cows. (**A**) Nuclear factor erythroid 2-related factor 2 (*Nrf2*); (**B**) superoxide dismutase (*SOD*); (**C**) cysteine uptake transporter (*XCT*); (**D**) hemeoxygenase 1 (*HO-1*); (**E**) NADPH-quinone oxidoreductase 1 (*NQO-1*). LCD: low-concentrate group, concentrate–forage = 4:6; HCD: high-concentrate group, concentrate–forage = 6:4; DAE: dandelion aqueous extract group, with the addition of 0.5% DAE in HCD diet. Different capital letters (A and B) indicate highly significant differences between treatments (*p <* 0.01). Different lowercase letters (a and b) indicate significant differences between treatments.

**Table 1 ijms-25-06075-t001:** Effects of different diets on milk quality of dairy cows.

Item	Treatment ^1^			SEM	*p*-Value
	LCD	HCD	DAE		
Week 1					
Butterfat (%)	4.06	3.23	3.51	0.366	0.091
Solid non-fat (%)	8.79	8.98	8.84	0.133	0.400
Intensity (kg/m^3^)	30.88 ^a^	32.60 ^b^	31.62 ^ab^	0.635	0.041
Protein (%)	3.43	3.50	3.45	0.049	0.379
Lactose (%)	4.98	5.08	5.00	0.075	0.451
Week 2					
Butterfat (%)	2.90	2.79	3.00	0.276	0.754
Solid non-fat (%)	8.32 ^A^	8.24 ^A^	8.59 ^B^	0.099	0.003
Intensity (kg/m^3^)	29.61 ^a^	30.05 ^ab^	30.87 ^b^	0.437	0.018
Protein (%)	3.22	3.25	3.29	0.045	0.269
Lactose (%)	4.72	4.70	4.84	0.062	0.058
Week 3					
Butterfat (%)	2.84	2.64	2.87	0.129	0.166
Solid non-fat (%)	8.39 ^A^	9.09 ^B^	8.94 ^B^	0.106	<0.001
Intensity (kg/m^3^)	29.93 ^A^	33.14 ^C^	32.01 ^B^	0.477	<0.001
Protein (%)	3.24 ^A^	3.29 ^A^	3.37 ^B^	0.035	0.002
Lactose (%)	4.77 ^A^	5.12 ^B^	5.06 ^B^	0.058	<0.001
Week 4					
Butterfat (%)	3.12	3.03	3.05	0.129	0.761
Solid non-fat (%)	8.65 ^a^	9.02 ^b^	8.96 ^B^	0.125	0.012
Intensity (kg/m^3)^	30.62 ^A^	32.49 ^B^	32.07 ^B^	0.453	0.001
Protein (%)	3.32	3.39	3.33	0.374	0.376
Lactose (%)	4.86 ^A^	5.11 ^B^	5.07 ^B^	0.071	0.003
Week 5					
Butterfat (%)	3.03 ^b^	2.67 ^a^	2.84 ^ab^	0.120	0.018
Solid non-fat (%)	8.70	8.86	8.70	0.109	0.309
Intensity (kg/m^3^)	31.11	31.95	31.35	0.358	0.069
Protein (%)	3.28	3.30	3.34	0.030	0.110
Lactose (%)	4.94	5.02	4.93	0.064	0.322

^1^ LCD: low-concentrate group, concentrate–forage = 4:6; HCD: high-concentrate group, concentrate–forage = 6:4; DAE: dandelion aqueous extract group, with the addition of 0.5% DAE in HCD diet. Different capital letters (A, B, and C) indicate highly significant differences between treatments (*p <* 0.01). Different lowercase letters (a and b) indicate significant differences between treatments.

**Table 2 ijms-25-06075-t002:** Ingredients and nutrient levels of different diets (dry matter basis).

Item	Treatment ^1^		
	LCD	HCD	DAE
Ingredient (% DM)			
Leymus chinensis	14.82	6.06	6.01
Corn silage	21.99	13.93	13.89
Alfalfa hay	12.70	10.97	10.92
Fresh distiller’s grains	5.85	3.61	3.64
Sugar beet meal	4.39	3.91	3.86
Cottonseed	0.00	1.73	1.74
Corn	21.79	32.37	32.20
Wheat bran	1.57	3.44	3.44
Palm oil	1.09	1.58	1.55
Soybean meal	5.48	7.74	7.73
Cottonseed meal	2.76	3.99	3.95
Distiller dried grain with solubles (DDGS)	3.88	5.63	5.59
Calcium hydrogen phosphate	1.13	1.61	1.57
Calcium carbonate (light powder)	0.62	0.93	0.95
Magnesium oxide	0.24	0.33	0.32
Salt	0.41	0.46	0.48
Vitamin and mineral premix ^2^	0.91	1.27	1.25
Urea premix	0.20	0.30	0.32
Yeast premix	0.17	0.14	0.13
Dandelion aqueous extracts (DAEs)	0.00	0.00	0.50
Concentrate–forage	4:6	6:4	6:4
Nutrient composition ^3^			
Net energy for lactation (NEL, MJ/Kg)	6.2	6.6	6.5
Crude protein (% DM)	15.4	16.1	16
Ether extract (% DM)	5.2	6	6
Neutral detergent fiber (% DM)	49.5	43.2	43
Acid detergent fiber (% DM)	22.5	18.9	18.8
Calcium (% DM)	1.04	1.04	1.03
Total phosphorus (% DM)	0.53	0.53	0.53

^1^ LCD: low-concentrate group, concentrate–forage = 4:6; HCD: high-concentrate group, concentrate–forage = 6:4; DAE: dandelion aqueous extract group, with the addition of 0.5% DAE in HCD diet. ^2^ The vitamin and mineral premix contained the following: vitamin A (540 × 10^3^ IU/kg), vitamin D3 (135 × 10^3^ IU/kg), vitamin E (2.70 × 10^3^ mg/kg), biotin (5.4 mg/kg), Mn (1.35 × 10^3^ mg/kg), Cu (1.35 × 10^3^ mg/kg), Zn (6.75 × 10^3^ mg/kg), I (90 mg/kg), Se (35 mg/kg), and Co (20 mg/kg). ^3^ The net energy for lactation (NEL, MJ/Kg) and the remaining nutrient composition (% DM) values.

**Table 3 ijms-25-06075-t003:** Primer sequences used for quantitative real-time PCR.

Gene	Primer Sequence	Product Size (bp)	GenBank Accession No.
*HO-1*	F: GGCAGCAAGGTGCAAGA	221	NM_001014912.1
	R: GAAGGAAGCCAGCCAAGAG		
*SOD*	F: GAGGCAAAGGGAGATACAGTC	197	NM_174615.2
	R: GTCACATTGCCCAGGTCTC		
*NQO-1*	F: GGTGCTCATAGGGGAGTTCG	235	NM_001034535.1
	R: GGGAGTGTGCCCAATGCTAT		
*XCT*	F: GATACAAACGCCCAGATATGC	136	XM_002694373.2
	R: ATGATGAAGCCAATCCCTGTA		
*GAPDH*	F: GGGTCATCATCTCTGCACCT	177	NM_001034034.2

*HO-1*: hemeoxygenase 1; *SOD*: superoxide dismutase; *NQO-1*: NADPH-quinone oxidoreductase 1; *XCT*: cysteine uptake transporter; *GAPDH*: glyceraldehyde-3-phosphate dehydrogenase.

## Data Availability

Data is contained within the article and Appendix A.

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
