# Peer review of "Effects of Dandelion Extract on Promoting Production Performance and Reducing Mammary Oxidative Stress in Dairy Cows Fed High-Concentrate Diet"

_ijms, 2024, doi:10.3390/ijms25116075_

Round 1

Reviewer 1 Report

Comments and Suggestions for Authors

1. Please provide the specific ethics approval number.

2. Why was 0.5% DAE chosen for supplementation, and how was this dosage determined?

3. The authors should note in the corresponding section of the methods that the HPLC results are provided in the supplementary materials.

4. Did the authors determine the amplification efficiency of the qPCR reaction before using the 2-∆∆CT method to determine gene expression? Please provide the amplification efficiency data for the qPCR reaction.

5. The author is advised to further validate the relevant genes through Western blot (WB) experiments.

6. The citations in the article are confusing.

7. Nrf2 can directly regulate the promoter activity of HO-1. Why did the DAE group significantly increase the expression of Nrf2 in this study, while the expression of HO-1 showed no significant difference compared to other groups?

Comments on the Quality of English Language

None

Author Response

Thank you very much for taking the time to review this manuscript. Please find the detailed responses below and the corresponding revisions highlighted in the re-submitted files. Comments and Suggestions for Authors 1. Please provide the specific ethics approval number. We have added the specific ethics approval number in line 503. 2. Why was 0.5% DAE chosen for supplementation, and how was this dosage determined? According to the previous study of in vivo experiments of cows with other plant extract added in the diet, and our previous in vitro study about the anti-oxidative effect of DAE on bovine mammary epithelial cells, the dosage of the extract were determined. 3. The authors should note in the corresponding section of the methods that the HPLC results are provided in the supplementary materials. Thank you for your reminder, and we have noted the supplementary figure in line 553. 4. Did the authors determine the amplification efficiency of the qPCR reaction before using the 2-∆∆CT method to determine gene expression? Please provide the amplification efficiency data for the qPCR reaction. Yes, we have already determined the amplification efficiency of the qPCR reaction, and we attached the original data in the supplemental file. And in the previous published paper from our team, we also use the same primer and amplification system. 5. The author is advised to further validate the relevant genes through Western blot (WB) experiments. Thank you so much for your suggestion, and we will consider the application of WB in our future studies. 6. The citations in the article are confusing. Sorry for the misleading about the citations, we have corrected the citation number and double checked the sequence. 7. Nrf2 can directly regulate the promoter activity of HO-1. Why did the DAE group significantly increase the expression of Nrf2 in this study, while the expression of HO-1 showed no significant difference compared to other groups? Thank you for your question, and that is also interesting for us. According to our previous study investigating anti-oxidative effects of DAE on bovine mammary epithelial cell lines (MAC-T cell) induced by LPS, HO-1 expression in MAC-T cells was significantly increased together with significant higher level of Nrf2 expression. And except for HO-1, the gene expression of SOD, XCT, and NQO-1 in DAE group were significantly increased compared with LPS group. For the current study, the results may be explained with different mechanisms responding to the increased expression of Nrf2 regulator in vivo compared with cell study. And in our further study, we will use different methods like WB and transcriptomics to explain this results.

Reviewer 2 Report

Comments and Suggestions for Authors

In the study entitled “Effects of dandelion extract in promoting production performance and reducing mammary oxidative stress in dairy cows fed high-concentrate diet” the Authors purpose was to examine the effects of rumen bypass dandelion extract on the lactation performance, immune index, and mammary oxidative stress of lactating dairy cows fed a high-concentrate diet.

According to their findings, Authors concluded that feeding a high-concentrated diet could increase the milk yield of dairy cows, but the milk quality, rumen homeostasis, and antioxidative capability were adversely affected. The supplementation of dandelion aqueous extract group (DAE) in a high-concentrate diet enhanced antioxidative capability by activating the Nrf2 regulatory factor and improved rumen homeostasis and production performance.

Please check the punctuation throughout the text as well as English language. On this regard, English language should be improved throughout the manuscript and several comments, reported below, should be solved.

Specific Comments

The title well reflects well the major findings of the study.

The abstract adequately summarize methodology, results, and significance of the study. However, Authors should specify the full-term of acronyms which appear for the first time. Moreover, the statistical analysis applied on the obtained data as well as the results together with P values should also indicated.   

The introduction section should be improved by adding some insights on the crucial physiological adaptation of dairy cow to peripartum and, more specifically, regarding the peculiarity of lactation phase in livestock. Moreover, in order to make stronger the rationale of the study, Authors should add more information concerning the breeding strategy studies focused on the improvement of animal reproduction and production and on the diet supplementation in veterinary field emphasizing the significant increase of interest showed by scientific community on diet improvement to enhance animal health status and welfare. Authors could add, at the beginning of introduction the following “Among the several crucial phases of productive cycle, the transition period lasting from the three weeks prior to three weeks after calving is considered as the most critical phase in mammals as they have to face with physiological body demands and adaptation. Particularly, lactation is a demanding period for the animal, which requires a significant effort of the organism to meet all needs, but also the effort of farmers to provide those needs. Despite the action of homeostatic mechanisms to maintain the nutritional balance, changes in metabolites and hormones occur as a result of increased metabolic demands in lactating animals. These changes make animals physiologically unstable and more susceptible to a number of metabolic diseases compromising productivity as well as welfare (Fiore E., et al., PLoS One, 2018, 13(4): e0193803; Fiore E., et al. 2014 Archiv Tierzucht 57, 1-9; Bazzano M. et al., (2014) Reproduction in domestic animals, 49 (6), pp. 947-953). Advances in reproduction research study enabled the identification of strategies to improve reproductive and productive capacity in livestock. The diet composition improvement represents a key factor to enhance the health status and welfare of animals as well as the products obtained from them (Monteverde V. et al., Journal of Applied Animal Research, 2017, 45: 615-618; Abbate J.M. et al., 2020, Animals, 10, 12, 1-13, 2303; Piccione G. et al., 2014, Journal of Equine Veterinary Science, Volume 34, Issue 10, Pages 1181 – 1187)."

In the section of Materials and Methods some clarifications are needed and some missing information should be added.

Did Authors evaluate the health status of enrolled animals? Have the enrolled animals been checked for parasitic infections? Overall, Authors should add more information on animals enrolled in the study.

Authors should indicate the respective “x g” of 3500 rpm.

Regarding ELISA kits used in the study, Authors should indicate whether the kits were specific for bovine species and/or whether them have been validated.

Moreover, Authors should indicate the sensitivity of the kits as well as the inter- and intra-variability.

Regarding statistical analysis, Authors wrote “The data were normalized using means ± standard errors (SEs).” This is unclear. Did the Authors apply a normality test on data in order to assess their normal distribution? Please clarify this aspect as Authors used a parametric analysis.

Results and Discussion section are well written and the findings obtained in the study were well presented, discussed and justified with appropriate references.

In the conclusion section Authors well summarize the results and the significance of the study. However, I suggest to add an introductory sentence.

Authors might check the citation throughout the text. They reported the ref as [i], [ii-v]; maybe Authors should indicate ref as [1], [2-5]. Please check and correct.

Comments on the Quality of English Language

English language should be improved throughout the manuscript by minor corrections. 

Author Response

Dear reviewer, thank you so much for your comments and suggestions, it help us a lot to improve our manuscript. And now I will response to your comments as listed below: 1. In the abstract, the full term of acronyms and P value were corrected and highlighted. 2. Based on your suggestions, we added some information at the beginning of introduction focus on the diet supplementation in veterinary field to enhance animal health status and welfare. 3. For the health status of enrolled animals, we firstly selected more than 200 cows from a dairy farm with similar lactating periods. Then we recorded the milk yield and measured somatic cell count before the study as the representatives of their production performance and health condition. Lastly, the 60 cows were selected based on their milk yield and somatic cell counts results. 4. For the speed of rotation, we followed the methods from some previous study and they used rpm instead of g. So can we still use rpm in our manuscirpt? 5. For the ELISA kits, they were specific for bovine, and we have already add these information in the materials and methods. For the sensitivity, each types of bovine ELISA kits have their specific and optimum range of detection, and we did pre-test and dilute our samples to meet the ranges. 6. For the statistical analysis, we use one way ANOVA in SPSS to do statistic analyze, which including homogeneity of variance test. So the data follow a normal distribution. I am sorry for the misleading of my language expression and I change the word from “normalized” to “presented”. 7. In the manuscript, we made some corrections to improve our English language with highlights.

Round 2

Reviewer 1 Report

Comments and Suggestions for Authors

None

Comments on the Quality of English Language

The writing of this article should be further polished